# Faster Distributed Synchronous SGD with Weak Synchronization

## Abstract

Distributed training of deep learning is widely conducted with large neural networks and large datasets. Besides asynchronous stochastic gradient descent (SGD), synchronous SGD is a reasonable alternative with better convergence guarantees. However, synchronous SGD suffers from stragglers. To make things worse, although there are some strategies dealing with slow workers, the issue of slow servers is commonly ignored. In this paper, we propose a new parameter server (PS) framework dealing with not only slow workers, but also slow servers by weakening the synchronization criterion. The empirical results show good performance when there are stragglers.

## 1 Introduction

Speed is crucial for training deep neural networks. Deep learning is now widely used in computer vision (Zeiler & Fergus, 2014; Simonyan & Zisserman, 2014; Szegedy et al., 2015; He et al., 2016), speech (Hinton et al., 2012a; Xiong et al., 2017), and natural language processing (Collobert et al., 2011; Wu et al., 2016). It benefits mainly from the complex representations, which require large neural networks and large datasets. While larger network architectures and larger datasets improve accuracy, they also require longer training times, which borders many researchers and developers. Hence, distributed training is a potential solution to improve the rate of updates.

Currently, mini-batch Stochastic Gradient Descent (SGD) and its variants are popular choices for training deep neural networks. For distributed training, it is common to use the Parameter Server (PS) architecture. It is composed of the server nodes and the worker nodes. The server nodes maintain a global copy of the parameters, receive the updates from the workers, apply the updates to the model, and broadcast the latest parameters to the workers. The worker nodes pull the latest parameters from the server nodes, perform the computation of the updates according to the local portion of the training data, and push the updates to the server nodes. The entire set of parameters is distributed to multiple server nodes. The full dataset and the corresponding workload is distributed to multiple worker nodes.

As shown in Figure 1, the updating mechanism of PS repeats the following 4 steps: (1). Workers pull the parameters from servers → (2). Workers compute the gradients → (3). Workers push the gradients to servers → (4). Servers aggregate the gradients and update the parameters. The updates can be synchronous or asynchronous. Synchronous training requires the aggregation of the gradients from all the workers before the updates are applied to the parameters in each iteration, while asynchronous training applies the updates whenever they are received by the server nodes. Note the Figure 1 also illustrates the architecture of communication between server nodes and worker nodes, which is a fully connected bipartite graph.

Typically, asynchronous SGD is more popular among practitioners because of faster and immediate updates. Without waiting for all the updates pushed by the workers, delays can be reduced. However, the asynchrony also brings more noise and higher variance to the updates (Liu et al., 2015), which can result in slow convergence or convergence to poor solutions.

Synchronous SGD can have faster and more stable convergence, but it suffers from waiting for synchronization. Servers must wait for all the workers to push their updates. Similarly, workers must wait for all the servers to send back the latest version of the parameters. To make things worse, the waiting time for synchronization increases when the number of nodes increases.

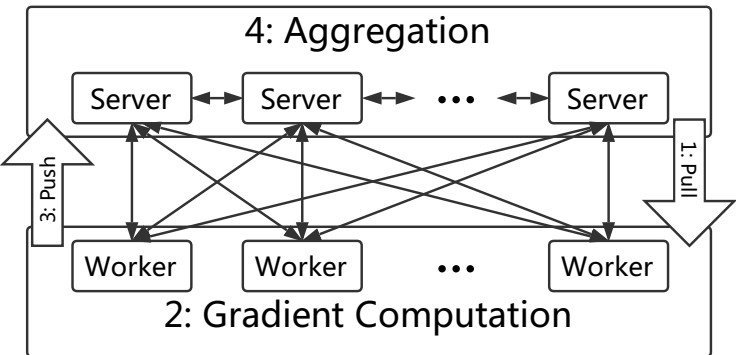

Figure 1: Parameter server architecture.

However, most of the previous research focuses on the issue of slow workers, while the servers can also be the stragglers. The communication between servers and workers in Figure 1 is mutual. The workers need servers' responses to continue the gradient computation. The servers can be slow due to networking issues such as unpredictable message delay, heterogeneous computational ability or memory capacity. A typical scenario which causes slow servers is that sometimes the server nodes and the worker nodes can be placed on the same machine, where the workers occupy most of the computational resources. Approaches like Stale Synchronous Parallel (SSP) (Dai et al., 2013; Ho et al., 2013; Li et al., 2014a;b) can be used to accelerate synchronous training. By upper-bounding the staleness between any two workers, the synchronization is weakened. As long as the bounded staleness criterion is satisfied, the servers can apply the updates whenever they are received by the server nodes. Waiting only happens when the bounded staleness criterion is not satisfied. Using backup workers (Pan et al., 2017) can also accelerate synchronous training.

In this paper, we propose a new PS framework, which deals with not only slow workers, but also slow servers by weakening the synchronization criterion. Any agent, regardless server or worker, can continue its own task even if the global pushing / pulling (i.e. steps 1 and 3 in Figure 1) in the current global iteration are not yet complete.

The main contributions of this paper are listed as follows:

- **New PS framework.** From the systems view, we implement a new PS framework that is tolerant to both slow workers and slow servers.

- **Partial synchronization in distributed training of deep neural networks.** The empirical results show that the proposed PS framework can accelerate the distributed training of deep neural networks with low validation errors.

- **Communication-efficiency.** The servers and workers can proactively choose not to send some of the messages to reduce communication overhead.

- **Illustration of the slow server effect.** We illustrate how the slow servers could influence the training speed, which has received scant attention in previous research.

## 2 SYNCHRONOUS SGD

The learning problem in this paper is minimizing the the loss function $L(w)$ defined as follows:

$$L(w) = \frac{1}{|\mathcal{X}|} \sum_{x \in \mathcal{X}} l(x, w), \tag{1}$$

where $\mathcal{X}$ is the dataset, $x \in \mathcal{X}$ is a data sample, $l(x, w)$ is the loss value computed by the data sample $x$ and the parameter (model) $w$.

In this paper, we use synchronous SGD as the basic algorithm to minimize the loss function (1), which can be formally described by the following update procedure:

$$\nabla \tilde{l}(w^t) = \frac{1}{|\mathcal{B}|} \sum_{x \in \mathcal{B}} \nabla l(x, w^t), \tag{2}$$

$$w^{t+1} = w^t - \eta^t Updater(\nabla \tilde{l}(w^t)), \tag{3}$$

where $t$ is the index of iteration, $\mathcal{B}$ is the minibatch, $\nabla \tilde{l}(w^t)$ is the aggregated gradient, $\eta^t$ is the learning rate, $Updater(\cdot)$ is the updater function which is used to compute the update of the parameters. The updater function can be any variant of gradient descent, such as momentum SGD (Qian, 1999), Nesterov accelerated SGD (Nesterov, 1983), Adam (Kingma & Ba, 2014), AdaGrad (Duchi et al., 2011), or RMSProp (Hinton et al., 2012b).

## 2.1 Distributed Synchronous SGD

The computation of the gradients in (2) can be distributed to multiple workers and then aggregated by the servers. To be more specific, (2) can be rewritten as

$$\nabla \tilde{l}(w^t) = \frac{1}{|\mathcal{B}|} \sum_{j=1}^{k} \sum_{x \in \mathcal{B}_j} \nabla l(x, w^t) = \frac{1}{kn} \sum_{j=1}^{k} \sum_{x \in \mathcal{B}_j} \nabla l(x, w^t),$$

where $j$ is the index of worker, $k$ is the total number of workers, $\mathcal{B}_j$ is the minibatch on the $j$th worker, and $\sum_j |\mathcal{B}_j| = |\mathcal{B}|$. We further assume that the minibatches of different workers all have the same size $n = |\mathcal{B}_j|$ for $\forall j \in [k]$.

Note that although from the point of view of each individual worker, the size of minibatch is $n$, for synchronous SGD, the actual size of the minibatch is $kn$.

In the distributed scenario, there could be multiple server nodes. Each server is responsible for storing and updating a portion (several blocks) of the entire set of parameters. Formally, the blockwise aggregation and updating is

$$\nabla \tilde{l}_i(w^t) = \frac{1}{|\mathcal{B}|} \sum_{j=1}^{k} \sum_{x \in \mathcal{B}_j} \nabla l_i(x, w^t) = \frac{1}{kn} \sum_{j=1}^{k} \sum_{x \in \mathcal{B}_j} \nabla l_i(x, w^t), \tag{4}$$

$$w_i^{t+1} = w_i^t - \eta^t Updater(\nabla \tilde{l}_i(w^t)), \tag{5}$$

where $\tilde{l}_i(w^t)$, $l_i(x, w^t)$, and $w_i^t$ are the $i$th blocks of $\tilde{l}(w^t)$, $l(x, w^t)$, and $w^t$.

For the basic synchronous SGD, the update (3) must wait until the aggregation (2) is finished. Thus, with larger $|\mathcal{B}|$, more gradients must be computed, which requires more time for each iteration. Although the distributed training can accelerate the computation of gradients, the communication and synchronization are often a bottleneck in practice. In some cases, distributed training can even be slower than single-node SGD. To deal with such issue, following two techniques are utilized.

## 2.2 Large Minibatch SGD

Goyal et al. (2017) empirically shows that distributed synchronous SGD can be accelerated by tuning the learning rate. To be more specific, fixing the original learning rate, when the size of minibatch is multiplied by $k$, the learning rate is also multiplied by $k$. Bottou et al. (2017) also theoretically discuss such linear scaling rule.

## 2.3 Partial Pushing

To deal with the issue of slow workers, Pan et al. (2017) introduced "backup workers" to the PS framework. In this paper, we call it "partial pushing" or "partial aggregation" because the server only waits for the first $m \leq k$ workers' before updating. All other delayed workers are simply ignored.

Integrated with blockwise aggregation on multiple servers, the partial pushing can be defined as

$$\nabla \tilde{l}_i^t(w^t) = \frac{1}{|\mathcal{A}^t|n} \sum_{j \in \mathcal{A}^t} \sum_{x \in \mathcal{B}_j} \nabla l_i(x, w^t), \tag{6}$$

$$w_i^{t+1} = w_i^t - \eta^t Updater(\nabla \tilde{l}_i^t(w^t)), \tag{7}$$

where $\mathcal{A}^t$ is the set of the indexes of the first $m$ responding workers in the $t$th iteration.

## 3 PROPOSED METHOD

As shown in Figure 1, before computing the gradient, each worker must finish the pulling, which requires receiving parameters from all the servers. When there are stragglers in the server group, all the other servers and workers will have to wait for the stragglers before starting the next iteration.

---

**Algorithm 1** Partially Synchronous Parallelism SGD (PSP-SGD)

---

**Server** $i = 1, 2, \ldots$**:**

---
1: Initialize $w_i^0$, set $t \leftarrow 0$
2: **for** $t = 0, 1, 2, \ldots$ **do**
3:     Broadcast parameters with timestamp $\langle w_i^t, t \rangle$ to all the workers
4:     $g \leftarrow 0, d \leftarrow 0$
5:     **repeat**
6:         Receive gradient block with timestamp $\langle g_i', t' \rangle$
7:         **if** $t' < t$ **then**
8:             Drop $g_i'$
9:         **else**
10:             $g_i \leftarrow g_i + g_i', d \leftarrow d + 1$
11:         **end if**
12:     **until** Timeout $\tau_1$ after $c$ gradients have been received or all workers have responded
13:     $w_i^{t+1} \leftarrow w_i^t - d\eta^t Updater(\frac{1}{d} g_i)$
14:     $t \leftarrow t + 1$
15: **end for**

**Worker** $i = 1, 2, \ldots$**:**

---
1: $t \leftarrow 0$
2: **for** $t = 0, 1, 2, \ldots$ **do**
3:     **if** $t > 0$ **then**
4:         $w^t \leftarrow w^{t-1}$
5:     **else**
6:         $w^t \leftarrow 0$
7:     **end if**
8:     **repeat**
9:         Receive the $j$th block and timestamp $\langle w_j^{t'}, t' \rangle$
10:         **if** $t' < t$ **then**
11:             Drop $w_j^{t'}$
12:         **else**
13:             $w_j^t \leftarrow w_j^{t'}$
14:         **end if**
15:     **until** Timeout $\tau_2$ after $b\%$ blocks have been received or all servers have responded
16:     Compute the gradient $g$ with $w^t$
17:     Push each block with timestamp $\langle g_j, t \rangle$ to its corresponding server
18:     $t \leftarrow t + 1$
19: **end for**

---

To deal with the issue of slow servers, we introduce *partial pulling*: in each iteration, a worker only waits for the first $b$ blocks rather than all the blocks of the parameters to arrive before computing the gradients. Similar to partial pushing, partial pulling also weakens the synchronization, which helps alleviate straggler effects. This is integrated with the partial pulling and learning rate tuning techniques introduced in Section 2. Furthermore, to make the algorithm more practical, we add a

timeout mechanism to both partial pushing and partial pulling. Combining all of these ideas results in a new algorithm which we call $\underline{P}$artially $\underline{S}$ynchronous $\underline{P}$arallelism SGD (PSP-SGD). The details are shown in Algorithm 1, where $c$ is the minimum number of aggregated pushes, $b$ is the minimum number of successful pulls, $w_i$ is the $i$th block of the parameter $w$, $g_i$ is the $i$th block of the gradient, $t$ is the timestamp, $d$ is the number of aggregated gradients, $\tau_1$ is the timeout threshold of partial pushing, and $\tau_2$ is the timeout threshold of partial pulling.

We summarize the techniques we use to accelerate synchronous SGD here:

- **Linear scaling rule.** In the server part of Algorithm 1, Line 13 uses the linear scaling rule introduced in Section 2.2 to tune the learning rate according to the number of aggregated gradients $d$.

- **Partial pushing.** In the server part of Algorithm 1, Line 7-12 uses the partial pushing introduced in Section 2.3. For each block of parameters, for the workers, at least $c$ gradients are guaranteed to be aggregated. After the minimum synchronization criterion are satisfied, the servers will wait up to $\tau_1$ of time before continuing to the next step.

- **Partial pulling.** In the worker part of Algorithm 1, Line 10-15 uses the partial pulling. For the workers, at least $b\%$ of the blocks are guaranteed to be pulled. After the minimum synchronization criterion are satisfied, the agents will wait up to $\tau_1$ or $\tau_2$ of time before continuing to the next step.

## 4 EXPERIMENTS

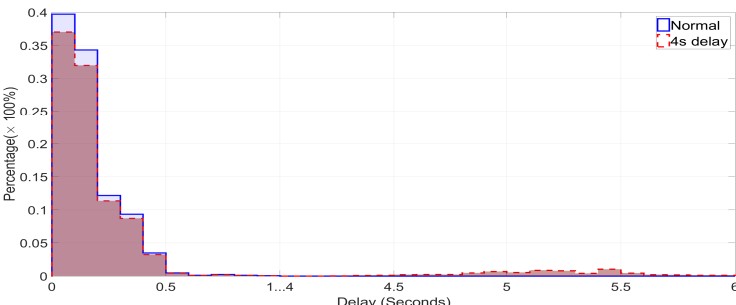

Figure 2: The PDF of message delay.

### 4.1 CLUSTER CONFIGURATION

In the distributed experiments, we use a cluster of 32 machines. Each machine has $4 \times 68$-core Intel Xeon Phi CPUs and 192GB RAM. The machines are connected with 1 Gbps Ethernet. All of the machines are used for placing both the server nodes and the worker nodes. We evenly distribute the servers and the workers to the other 32 machines.

Table 1: Numerical results with simulated delays, $c$ is the threshold of partial pushing, $b$ is the threshold of partial pulling

| Partial pushing with $c$ equals to | 32 | 28 | 28 | 20 |
|---|---|---|---|---|
| Partial pulling with $b$ equals to | 100% | 100% | 90% | 75% |
| Running time | 199.97 | 164.97 | 139.95 | 136.54 |
| Top-1 validation error | 0.1479 | 0.1609 | 0.1565 | 0.2914 |

### 4.2 EXPERIMENT CONFIGURATION

We conduct experiments on the ResNet-50 (He et al., 2016) model trained on CIFAR-10 (Krizhevsky & Hinton, 2009) dataset. Our implementation is based on MXNET (Chen et al., 2015). We run 90 epochs for each experiment. The learning rate is set as $0.02 \times \frac{dn}{128}$, where $d$ is the number of responding workers, $n$ is the minibatch size of a single worker. The learning rate will decrease by a

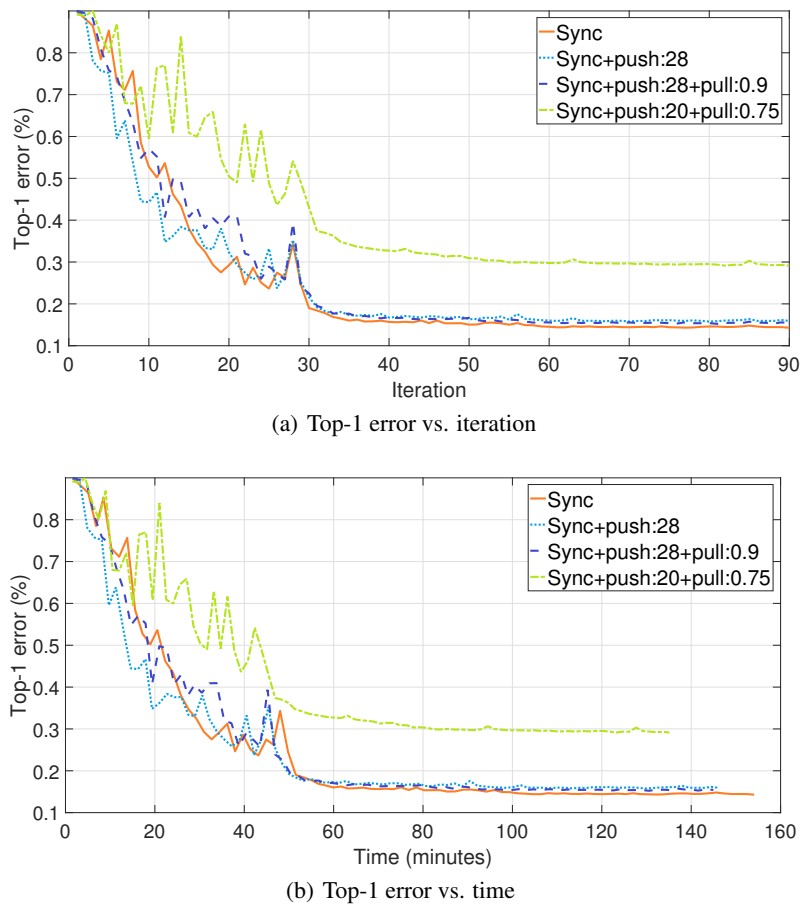

(a) Top-1 error vs. iteration

(b) Top-1 error vs. time

Figure 3: Top-1 validation error on CIFAR-10. With appropriate hyperparameters, using partial pushing and partial pulling, the algorithms can still have low validation error and converge faster at the same time

factor of $0.1$ at the 30th and 60th epochs. We also use a weight decay of $0.0001$ and momentum of $0.9$. We use top-1 validation error to measure the performance.

We use the naive synchronous SGD and partial pushing as the baselines. For each experiment, we launch 32 servers and 32 workers. The size of minibatch for each worker is $n = 160$. Thus, the actual size of minibatch for each iteration is $160 \times 32 = 5120$. The naive synchronous SGD is denoted as *Sync*. For synchronous SGD with partial pushing, the threshold $c$ is set as $28$, which means the servers only need to wait 28 instead of 32 workers to respond. This setting is denoted as *Sync+push:28*. For the proposed method, which is synchronous SGD with partial pushing and partial pulling, we test 2 different settings: (i). the servers wait for the first $c = 28$ pushing and the workers wait for the first $b = 90\%$ pulling, which is denoted as *Sync+push:28+pull:0.9*; (ii). the servers wait for the first $c = 20$ pushing and the workers wait for the first $b = 75\%$ pulling, which is denoted as *Sync+push:20+pull:0.75*. We run each experiment 3 times and take the average. Note that in these experiments, we do not use the timeout mechanism, which means we set $\tau_1 = 0$ and $\tau_2 = 0$ in Algorithm 1.

Besides the normal scenario, we also test the proposed algorithm with simulated message delay to show the efficiency. The general idea of the simulation is that the delays are rare but large. We randomly pick $0.16\%$ of the responses of pulling to have extra delays of $4$ seconds, which makes some servers stragglers.

In Figure 2, we plot the probability density function (PDF) of the the delays of pulling in the normal scenario and the scenario with simulated extra delays. The delay is measured by the duration between the times when the pulling request is sent and the time when the response arrives.

### 4.3 EMPIRICAL RESULTS

In Figure 3, we illustrate the performance of our algorithm in the normal scenario without simulated delays. We can see that with appropriate hyperparameters, using partial pushing and partial pulling, the algorithms can still have low validation error and converge faster at the same time.

In Figure 4, we illustrate the performance of our algorithm with simulated delays. We observe that with appropriate hyperparameters, with partial pushing and partial pulling, the algorithms can remain as effective as when there is no extra delay at all as expected, which reduces the training time. In Table 1, we list the numerical results of the experiments. It is shown that with partial pushing only, the training time can be reduced by nearly 17% compared to the naive synchronous SGD. Furthermore, by using both partial pushing and partial pulling with appropriate hyperparameters (i.e. $c = 28$, $b = 90\%$), the training time can be reduced by nearly 30%, with the validation error remaining low. However, when we set $c$ and $b$ to be too small, the algorithm may end up with a poor solution with relatively high validation error.

We conclude that the proposed method can efficiently deal with the slow servers with appropriate choice of hyperparameters $c$ and $b$. When there are no severely slow servers, our algorithm performs as good as the baseline. When there are slow servers, our algorithm can train the model much faster than the baseline with almost the same validation accuracy.

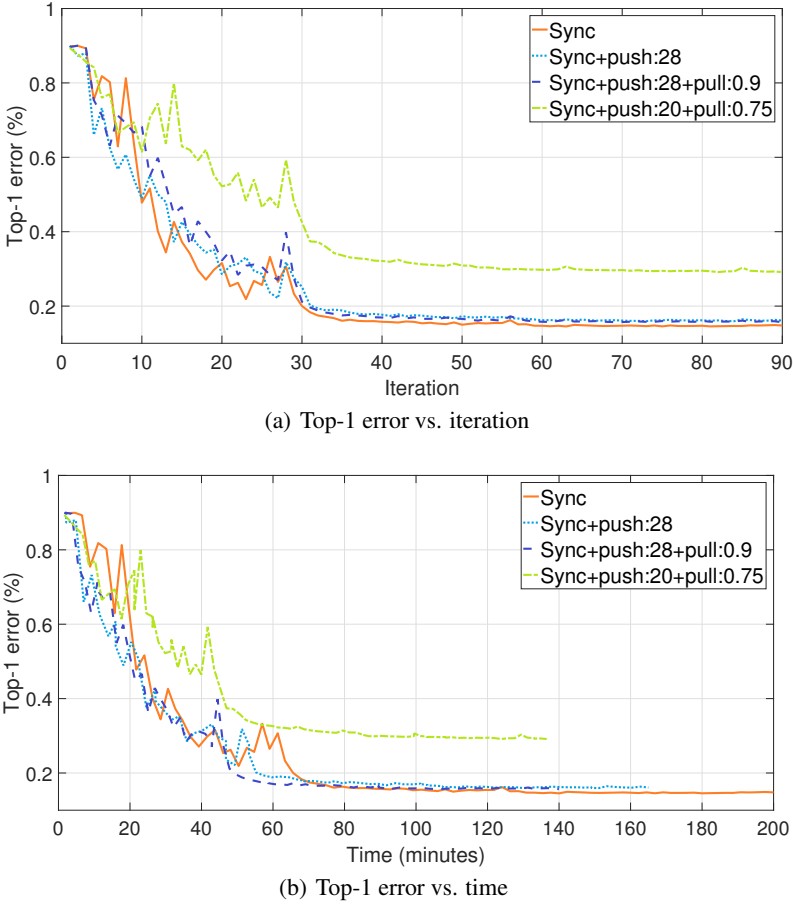

(a) Top-1 error vs. iteration

(b) Top-1 error vs. time

Figure 4: Top-1 validation error on CIFAR-10 with Simulated Delay. Compared to the naive synchronous SGD, training time can be reduced by nearly 17% using partial pushing, and reduced by nearly 30% using partial pushing and partial pulling.

### 4.4 DISCUSSION

The experiments show the efficiency of the proposed algorithm. The partial pulling can deal with slow servers, accelerate the training , while leaving the validation accuracy unharmed. It is shown that the synchronous SGD can be be very slow when a very small portion of messages are delayed. In our simulation, the naive synchronous SGD is prolonged by nearly 30%. Actually, with appropriate value of $c$ and $b$, the training time can be almost the same as the case with is no delayed messages at all, while also leaving the validation accuracy unharmed. However, users should be careful when using small $c$ and $b$, which will incur additional noise to the updates and may result in poor solutions. Note that we do not use the timeout mechanism in the experiments to make the illustrations clearer. Using timeout, with larger $\tau_1$ and $\tau_2$, the performance will be closer to naive synchronous SGD.

## 5 RELATED WORK

**Stale Synchronous Parallelism**    The PS frameworks with SSP such as Parameter Server Li et al. (2014b;a) and Petuum Dai et al. (2013); Ho et al. (2013) focus on solving the issues of slow workers. Such idea is orthogonal to this paper. As SSP can be view as another way to weaken the synchronization for the pushing, it can be potentially combined into our algorithm by replacing the partial pushing with SSP. The efficiency of such combination requires future research.

**Asynchronous SGD**    Asynchronous SGD is an widely-used alternative to synchronous SGD, whose synchronization criterion is fully relaxed. There are many variants of asynchronous SGD, such as Nesterov accelerated asynchronous SGD with variance reduction (Meng et al., 2016), and asynchronous SGD with delay compensation (Zheng et al., 2017). Although SSP and synchronous SGD have better theoretical results compared to asynchronous SGD, it is still unclear that which one is better in practice.

## 6 CONCLUSION

In this paper, we study the issue of slow servers for synchronous SGD. We propose a new parameter server (PS) framework dealing with not only slow workers, but also slow servers by weakening the synchronization criterion. The empirical results show good performance when there are stragglers.

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
