# OpenReview forum: "Faster Distributed Synchronous SGD with Weak Synchronization"
_ICLR.cc/2018/Conference — Reject_

### Official Review · AnonReviewer1 · 2017-11-27
**Need to motivate the problem and evaluate thoroughly**

**Rating:** 4
**Confidence:** 5

**Review:**

This paper introduces a parameter server architecture to improve distributed training of CNNs in the presence of stragglers. Specifically, the paper proposes partial pulling where a worker only waits for first b blocks rather than all the blocks of the parameters. This technique is combined with existing methods such as partial pushing (Pan et. al. 2017) for a partial synchronous SGD method. The method is evaluated with Resnet -50 using synthetic delays.

Comments for the author:

The paper is well-written and easy to follow. The problem of synchronization costs being addressed is important but it is unclear how much of this is arising due to large blocks.

1) The partial pushing method (Pan et. al. 2017, section 3.1) shows a clear evidence for the problem using a real workload with a large number of workers. Unfortunately, in your Figure 2, this is not as obvious and not real since it is using simulated delays. More specifically, it is not clear how the workers behave in a real environment and whether you get a clear benefit from using a partial number of blocks as opposed to sending all of them.

2) Did you modify your code to support block-wise sending of gradients (some description of how the framework was modified will be helpful)? The idea is to send partial parameter blocks and when 'b' blocks are received, compute the gradients. I feel that, with such a design, you may actually end up hurting the performance by sending a large number of small packets in the no failure case. For real, large data centers, this may cause a packet storm and subsequent throughput collapse (e.g. the incast problem). You need to show the evidence that you do not hurt the failure-free case for a large number of workers.

3) The evaluation is on fairly small workloads (CIFAR-10). Again, evaluating over Imagenet and demonstrating a clear speedup over existing sync methods will be helpful. Furthermore, a clear description of your “pull” configuration (such as in Figure 1) i.e. how many actual bytes or blocks are sent and what is the threshold will be helpful (beyond a vague 90%).

4) Another concern with partial synchronization methods that I have is that how do you pick these configurations (pull 0.75 etc). These appear to be dataset specific and finding the optimal configuration here requires significant experimentation that takes significantly more time than just running the baseline.

Overall, I feel there is not enough evidence for the problem specifically generating large blocks of gradients and this needs to be clearly shown. To propose a solution for stragglers, evaluation should be done in a datacenter environment with the presence of stragglers (and not small workloads with synthetic delays). Furthermore, the proposed technique despite the simplicity appears as a rather incremental contribution.

---

### Official Review · AnonReviewer2 · 2017-11-27
**Overall, the proposed method is not well-motivated, simple, with no theoretical support, and experimental results are not convincing.**

**Rating:** 3
**Confidence:** 4

**Review:**

This paper considers distributed synchronous SGD, and proposes to use "partial pulling" to alleviate the problem with slow servers.

The motivation is that the server may be a straggler. The authors suggested one possibility, namely that the server and some workers are located on the same machine and the workers take most of the computational resource. However, if this is the case, a simple solution would be to move the server to a different node. A more convincing argument for a slow server should be provided.

Though the authors claimed that they used 3 techniques to accelerate synchronous SGD, only partial pulling is proposed by them (the other 2 are borrowed straightforwardly from existing papers). The mechanism of partial pulling is very simple (just let SGD proceed after pulling a partial parameter block instead of the whole block). As mentioned by the authors in section 1, any relaxation in synchrony brings more noise and higher variance to the updates, and also may cause slow convergence or convergence to a poor solution. However, the authors provided no theoretical study on any of these aspects.

Experimental results are not convincing. Only one relatively small dataset (cifar10) is used Moreover, the slow server problem is only simulated by artificially adding delays to the server.

---

### Official Review · AnonReviewer3 · 2017-11-27
**Weak synchronization with a weak impact**

**Rating:** 4
**Confidence:** 5

**Review:**

Paper proposes a weak synchronization approach to synchronous SGD with the goal of improving even with slow parameter servers. This is an improvement on earlier proposals (e.g. Revisiting Synchronous SGD) that allow for slow workers. Empirical results on ResNet50 on CIFAR show promising results for simulations with slow workers and servers, with the proposed approach.

Issues with the paper:
- Since the paper is focused on empirical results, having results only for ResNet50 on CIFAR is very limiting
- Empirical results are based on simulations and not real workloads. The choice of simulation constants (% delayed, and delay time) seems somewhat arbitrary as well.
- For the simulated results, the comparisons seem unfair since the validation error is different. It will be useful to also provide time to a certain accuracy that all of them get to e.g. the validation error of 0.1609 (reached by the 3 important cases).

Overall, the paper proposes an interesting improvement to this area of synchronous training, however it is unable to validate the impact of this proposal.

---

### Decision · Program_Chairs · 2018-01-29
**ICLR 2018 Conference Acceptance Decision**

**Decision:**

Reject

**Comment:**

This paper introduces a method for making synchronous SGD more resistant to failed or slow workers. The idea seems plausible, but as the reviewers point out, the novelty and the experimental validation are somewhat limited. For a contribution such as this, it would be good to see some experiments on a wider range of tasks, and experiments with real rather than simulated workloads. I don't think this work is ready for publication at ICLR.